# High-resolution physical-biogeochemical structure of a filament and an eddy of upwelled water off Northwest Africa

Wilken-Jon von Appen[1], Volker H. Strass[1], Astrid Bracher[1], Hongyan Xi[1], Cora Hörstmann[1], Morten H. Iversen[1], and Anya M. Waite[2]

[1]Alfred Wegener Institute Helmholtz Centre for Polar and Marine Research, Am Handelshafen 12, 27570 Bremerhaven, Germany
[2]Department of Oceanography, and the Ocean Frontier Institute, Dalhousie University, Halifax, Nova Scotia, Canada

**Correspondence:** Wilken-Jon von Appen (Wilken-Jon.von.Appen@awi.de)

**Abstract.** Nutrient rich water upwells offshore of Northwest Africa and is subsequently advected westwards. There it forms eddies and filaments with a rich spatial structure of physical and biological/biogeochemical properties. Here we present a high resolution (2.5 km) section through upwelling filaments and an eddy obtained in May 2018 with a Triaxus towed vehicle equipped with various oceanographic sensors. Physical processes at the mesoscale and submesoscale such as symmetric insta-
5 bility, trapping of fluid in eddies, and subduction of low potential vorticity (which we use as a water mass tracer) water can explain the observed distribution of biological production and export. We found a nitrate excess (higher concentrations of nitrate than expected from oxygen values if only influenced by production and remineralization processes) core of an anti-cyclonic mode water eddy. We also found a high nitrate concentration region of ≈5 km width in the mixed layer where symmetric instability appears to have injected nutrients from below into the euphotic zone. Similarly further south high chlorophyll-*a*
10 concentrations suggest that nutrients had been injected there a few days earlier. Considering that such interactions of physics and biology are ubiquitous in the world's upwelling regions, we assume that they strongly influence the productivity of such systems and their role in $CO_2$ uptake. The intricate interplay of different parameters at kilometer scale needs to be taken into account when interpreting single profile and/or bottle data in dynamically active regions of the ocean.

January 24, 2020

# 1 Introduction

In the euphotic zone of a stratified ocean, productivity tends to be limited by nutrients as vertical exchange is suppressed in geostrophically balanced flows. However, dynamic processes associated with fronts may inject nutrient rich water from below into the mixed layer which enhances productivity (Strass, 1992) at scales that are difficult to quantify. The large upwelling systems of the world are located on the eastern side of the world's oceans (Chavez and Messié, 2009) such as the Mauretanian Upwelling system offshore of Northwest Africa. There the wind stress is such (northerly in the northern hemisphere) that cold nutrient rich water from depth is upwelled to the surface at the coast resulting in high productivity. In Eastern Boundary Upwelling Systems (EBUS), the upwelled water near the coast is separated by a sharp front from the often nutrient depleted water offshore (Capet et al., 2008a). This front is typically baroclinically unstable and can form eddies as well as mesoscale filaments, which may efficiently transport upwelled water offshore for hundreds of kilometers (Hosegood et al., 2017). This detachment of filaments is often related to topographic features along the coast line (Meunier et al., 2012).

At geostrophically balanced (i.e. low Rossby number meaning that the horizontal scales of motion are so large that Earth's rotation is dominant in the force balance) mesoscale filaments, cross-front exchange is weak and prohibits the resupply of nutrients to the euphotic zone in the filaments. As a result, new production in a mesoscale upwelling filament is limited as it depends on the initial injection of nutrients during the upwelling process (Hosegood et al., 2017). For larger Rossby numbers (i.e. approaching 1), the pure 2D geostrophic balance breaks down and 3D flows arise that contain a significant ageostrophic component which can lead to cross-front fluxes of e.g. water, mass, and nutrients. This submesoscale flow regime can involve vertical velocities on the order of 100 m/day. This ostensibly supplies nutrients to the mixed layer at fronts (Lévy et al., 2001, 2012) and filaments, especially at filaments generated from EBUS that propagate into nutrient depleted waters located offshore (Capet et al., 2008b). Submesoscale motions can also rapidly restratify the upper ocean in the presence of horizontal density gradients (Mahadevan et al., 2012).

Demonstrating the importance of the nutrient resupply mechanism, Hosegood et al. (2017) found that submesoscale dynamics via vertical circulations could explain a higher level of productivity in an upwelling filament off Northwest Africa than could be supported only by the initial nutrient supply during the coastal upwelling. In addition to the continuous input of nutrients from below the mixed layer, nitrogen regeneration may support new production in the filaments (Clark et al., 2016). Such fronts and filaments are ubiquitous over the world's oceans, especially in regions that have processes which generate lateral density gradients, but their overall impact on vertical nutrient supply remains understudied (Mahadevan, 2016).

Here we investigate a system of submesoscale filaments and a mesoscale eddy in the Cape Verde Frontal Zone between 20°N and 26°N off the Northwest African coast. There isopycnal mixing between two water masses takes place (Tomczak, 1981; Martínez-Marrero et al., 2008). These water masses are the fresher South Atlantic Central Water (SACW) defined along a line from 7.24°C/34.95 to 16.00°C/35.77 in TS space and the saltier North Atlantic Central Water (NACW) defined along a line from 7.50°C/35.05 via 11.00°C/35.47 to 18.65°C/36.76 in TS space (Tomczak, 1981). The salinity contrast of the two central waters stems from the excess evaporation in the subtropical North Atlantic (NACW) while the low salinity Antarctic Intermediate Water affects the South Atlantic (SACW).

Offshore of Mauretania, along the coast of Northwest Africa, the offshore motion of the upwelled water was studied in 2009 (Hosegood et al., 2017). Sea surface temperature (SST) showed that upwelling of cold and nutrient rich water close to the coast produced a strong bloom. This water was subsequently advected offshore and formed a filament that moved out into the ocean towards the west. A number of sections across the filament's edge showed strong temperature and salinity gradients between the surface ocean and greater depth. Upward movement of water appeared to take place within the filament caused by strong

horizontal velocity gradients (measured by the Rossby number $Ro = \frac{\zeta}{f}$ where $\zeta = \frac{\partial v}{\partial x} - \frac{\partial u}{\partial y}$ is the relative vorticity and $f$ is the planetary vorticity). Potential vorticity $PV \approx N^2(f + \zeta)$ (where $N^2$ is the buoyancy frequency indicating the strength of the stratification) values of opposite sign to $f$ in an otherwise gravitationally stable upper ocean, i.e. $\zeta/f < -1$, are a necessary condition for symmetric instability to occur (Haine and Marshall, 1998). This instability leads to slantwise convection whereby water parcels are exchanged in the vertical and horizontal at the same time. This exchange can be an efficient way to bring water

from below the mixed layer into the mixed layer associated with small scale mixed layer eddies and filaments characterized by $Ro \gtrsim 1$ and outcropping isopycnals. At such locations, Hosegood et al. (2017) found higher phytoplankton concentrations (as shown by chlorophyll-$a$ fluorescence).

     Eddies are another important oceanographic feature impacting vertical nutrient fluxes. They are often classified according to their rotation where cyclonic (anti-cyclonic) eddies are associated with upward (downward) displaced isopycnals in the eddy

center compared to the surrounding water. However, there are also eddies that contain downward displaced isopycnals below upward displaced ones. As the depth interval in-between creates a volume of low stratification (i.e. nearly of homogenous properties) called a "mode", these hybrid eddies are often called anticyclonic modewater eddies (ACME) or intra-thermocline eddies (McWilliams, 1985; Thomas, 2008). In the eastern tropical North Atlantic ACMEs make up an estimated 9% of all eddies (Schütte et al., 2016). Anti-cyclonic eddies with trapped fluid (translational velocity less than peak azimuthal velocity)

are retention regions of biogeochemical properties (d'Ovidio et al., 2013). The relative vorticity associated with anti-cyclonic eddies modifies the propagation of near inertial internal waves; their energy propagates downward and part of their energy is dissipated to vertical mixing (Kunze et al., 1995) at certain depths. The dissipation region for ACMEs is below their cores (Lee and Niiler, 1998).

     If frictional processes can be neglected, potential vorticity (PV) is conserved in the ocean below the mixed layer. Wind

forcing at the surface can drive PV to zero (Thomas, 2005). Therefore, upwelled water from EBUSs which has moved westward has low PV which—when subducted into eddies—results in low stratification cores of anti-cyclonically rotating eddies. An anti-cyclonic mode water eddy that travelled westwards from the upwelling region to Cape Verde some 800 km away was measured by several glider sections in 2014 (Karstensen et al., 2017). The eddy contained a lens of low salinity water in its center and had lowered isopycnals below this lens and slightly raised isopycnals above. This lens corresponded to a maximum

in nitrate and almost completely depleted oxygen. Karstensen et al. (2017) suggested a mechanism whereby, as the eddy propagates, at its rim nutrients are resupplied into the mixed layer through small scale motions. In the mixed layer those nutrients move inwards and then the organic matter sinks down and gets remineralized in the core. This leads to the high nitrate concentrations in the core with a corresponding near complete consumption of oxygen. In other words, through the gravitational settling of organic matter, the tracers nitrate and oxygen may evolve in a different way than a simple anti-correlation.

Eddies may also funnel sinking organic matter on its way to the sea floor. As particles sink in an anti-cyclone, ageostrophic velocities associated with the eddy may deflect the particles towards the center of the eddy resulting in a so-called "wine glass shape" in the particle distribution (Waite et al., 2016).

Biological-physical coupling at kilometer scale has rarely been observed in the ocean due to observational challenges. Specifically, here we ask what aspects of highly resolved observed biological structures can be explained by hydrography

(water masses) or physical dynamics (velocities) versus active biological processes. Here we present a high-resolution section through a filament and an eddy of upwelled water offshore of Northwest Africa that reveals many of the above discussed processes in a hitherto unavailable detail. In Section 2 we describe the used data. Section 3 then presents (in this order) physical, biogeochemical, and biological results and discusses them. Our conclusions are in Section 4.

## 2  Data

We used a MacArtney Triaxus E (extended version) towed undulating system which was towed at ≈8 knots behind RV *Polarstern*. The Triaxus flew a so-called saw-tooth pattern (Figure 1a) and was slightly deflected to the side by yaw flaps so as not to measure in the ship's wake. Here we occupied a northnortheastward track (Figure 2). The vehicle moved from 3–4 m below the surface to 350 m before coming back up to 3–4 m. At ≈1 m/s vertical and 8 knots≈4 m/s horizontal speeds this results in a ≈2.5 km distance between consecutive downcasts. Using both down- and upcasts, one could in principle get an even

higher—though non-constant—horizontal resolution. Over the total section distance of 278 km, 110 down- and upcasts were performed (Figure 1b) resulting in a very dense (less than the first baroclinic Rossby radius of ≈45 km in the region, Chelton et al., 1998) spatial coverage. The section was occupied between 30-May-2018 17:30 UTC and 31-May-2018 16:00 UTC. It was part of RV *Polarstern* cruise PS113 (Strass, 2018) that went from Chile to Germany; an overview of the encountered oceanography is given in Leach et al. (in prep). As described below, in addition to the Triaxus, we also used vessel mounted

sensor data, analyzed underway samples, and interpreted satellite data.

### 2.1  Triaxus

The Triaxus is 1.95 m wide, 1.25 m tall and 1.85 m long and in the configuration used it had the following sensors: Seabird SBE911+ with dual temperature/conductivity sensors and the following auxiliary sensors: SBE43 dissolved oxygen, SBE18 pH, WETLabs C-Star transmissiometer, WETLabs WETStar Environmental Characterization Optics (ECO) fluorometer, Sat-

lantic Biospherical photosynthetically available radiation (PAR). Furthermore, the Triaxus contained a Satlantic Deep SUNA nitrate sensor, and a TriOS RAMSES hyperspectral irradiance sensor.. The sensor meta data is available at https://hdl.handle.net/10013/sensor.5c126f5b-86de-469c-adf7-251789e54362 and the repository of the raw data is von Appen et al. (2019).

The SBE911+ and the auxiliary sensor data were processed using standard Seabird routines. The deviation between the two temperature/conductivity pairs was monitored throughout the cruise and no events of changing differences were detected.

The fluorescence from the ECO fluorometer was converted to chlorophyll-*a* concentrations using the factory values with-

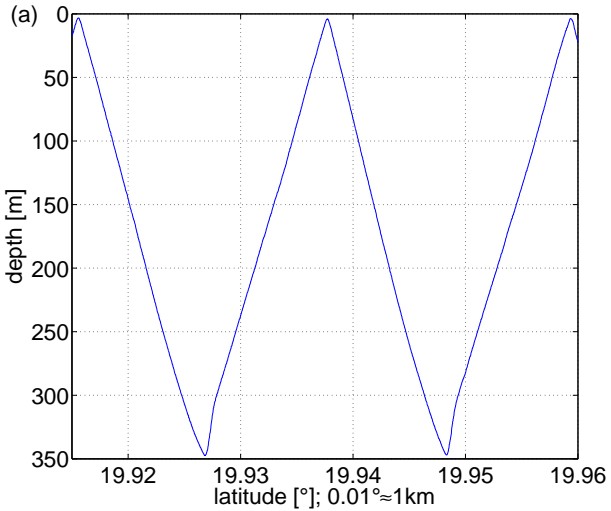 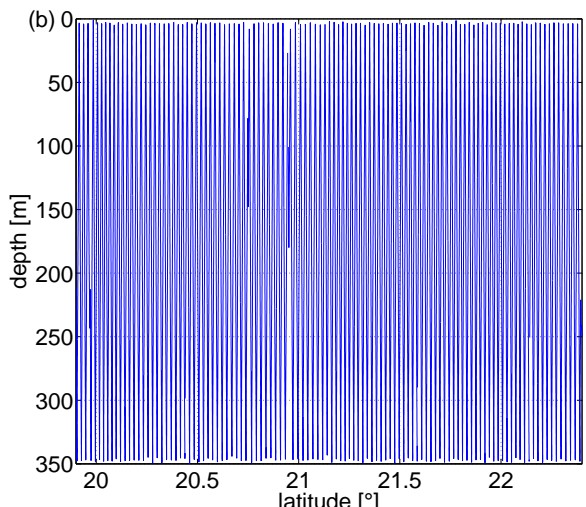

**Figure 1.** Saw tooth pattern of observations: (a) shows an exemplary subset of the down- and upcasts and (b) shows all 110 used undulations of the section from 30-May-2018 17:30 UTC to 31-May-2018 16:00 UTC. The distance between consecutive downcasts is ≈2.5 km.

out any further in-situ calibrations. Attenuation was calculated as $-1/(0.25$ m)*log10((transmissivity$-2.5\%)/100\%)$ where $100\%-2.5\%=97.5\%$ corresponded to the maximum transmissivity at depth ($>250$ m) which we presume to be a sensor offset.

The Deep SUNA was processed using Seabird UCI Version 1.2.1 in temperature/salinity (taken from the CTD) correction mode. At sea, nitrate standard solutions of 7 $\mu$mol/l and 14 $\mu$mol/l nitrate were used to obtain calibration point values. For this
purpose, the nitrate standard of Merck Millipore Article No.: 1.19811.0500 was diluted accordingly with water and 35 g/l NaCl and 0.5 g/l NaHCO$_3$. From the combination of these, the manufacturer suggests that one can achieve accuracies better than 2 $\mu$mol/l. A comparison to underway samples and CTD bottle calibration samples showed that they are in excellent qualitative agreement and a correction value of minus 1 $\mu$mol/l was determined. Especially where the samples showed values below 0.15 $\mu$mol/l (21.5–22.5°N), the sensor showed very constant values around 1 $\mu$mol/l. We therefore subtracted 1 $\mu$mol/l from
the sensor values and set the few values below 0 $\mu$mol/l to zero given that negative concentrations are not possible.

Near the upper and lower turning points of the saw tooth pattern (see Figure 1), the vertical property gradients in the ocean are traversed in opposite directions (upwards and downwards) within short periods of time. Since the gradients (at least on average) will not have changed over that time period, the upcast and downcast data should be identical. For temperature, this is roughly the case, but for other sensors it is not due to their sensor lag. Attempting to shift the sensor data in time with
respect to temperature such that the gradients, on average, become identical allows for a determination of the sensor lag. For oxygen we determined a sensor lag with respect to the temperature sensor of 2.5 s and for nitrate we determined a lag with respect to the temperature sensor of 20 s (though this may also be partially due to the separate data acquisition hardware of the nitrate sensor). Both sensor lags were corrected for. To further reduce inconsistencies from sensor lags, we only used data from downcasts thereby reducing our horizontal resolution to ≈2.5 km.

During the cruise, the traditional CTD rosette (Strass, 2019) was also deployed on average once a day. For the section presented here, the CTD rosette was deployed immediately before deployment and immediately after recovery of the Triaxus. Water samples from 6 depths were analyzed for inorganic nutrients (except for ammonium) (see below). We compared the properties measured by the rosette CTD and the Triaxus CTD in physical and TS space and did not detect systematic deviations. Furthermore, we emphasize that our goal here was not to describe highly accurate deep ocean parameters, but rather relative differences in oceanic parameters in the highly variable upper 350 m of the water column. This therefore does not require as much calibration effort as would typically be done for full ocean depth CTD rosette casts.

Hyperspectral downwelling irradiance at wavelength $\lambda$ from 320 nm to 950 nm with an optical resolution of 3.3 nm and a spectral accuracy of 0.3 nm at depth $z$, $E_d(\lambda, z)$ down to the maximum light depth (mostly $\lesssim 0.1\%$ of surface light) was measured by two identical irradiance radiometers (RAMSES ACC-2-VIS, TriOS GmbH, Germany). One was installed on a frame and lowered off the side of the ship before the CTD stations and the other one was installed on the Triaxus system. $E_d(\lambda, z)$ data were further processed to the mean spectral diffuse attenuation coefficient for the upper 30 m, $k_{d,mean}(\lambda)$, including incident sunlight correction and correction for surface water waves following Mueller et al. (2003) and Stramski et al. (2008), as detailed in Taylor et al. (2011). To obtain the chlorophyll-*a* of seven major phytoplankton groups (diatoms, dinoflagellates, haptophytes, *Prochlorococcus*, cyanobacteria excluding *Prochlorococcus*, chlorophytes, and chrysophytes) within the upper 30 m from the radiometry data, we followed the method of Bracher et al. (2015). Instead of using spectral remote sensing reflectance as in Bracher et al. (2015), we applied the empirical orthogonal function (EOF) analysis to our $k_{d,mean}(\lambda)$ data set. Subsequently we developed multiple linear regression models with the HPLC pigment (see underway samples subsection below) based phytoplankton group chlorophyll-*a* (instead of using the phytoplankton pigment concentrations as in Bracher et al., 2015) as the response variable and EOF loadings as predictor variables. The model to predict seven phytoplankton groups was relatively robust. More details on the data processing, model construction and cross validation results can be found in Bracher et al. (in review). Due to the passive nature of the sensor, the analysis is only available during day time. Here we only focus on the average over the top 30 m of the water column; vertical profiles of the phytoplankton groups along the saw-tooth pattern of our section are presented in Bracher et al. (in review).

In addition to the setup used here, the Triaxus contains a 1200 kHz upward and a 1200 kHz downward looking ADCP for the estimation of turbulence and the ocean currents at higher vertical resolution as well as an EK80 echo sounder for the estimation of zooplankton and fish biomass. With a GAPS ultra-short base line system its position can be tracked exactly. Furthermore, the Triaxus contains a Wetlabs AC-S spectral absorption and attenuation sensor which will in the future allow for phytoplankton group determination also during night time.

Laplacian splines under tension (Smith and Wessel, 1990) were used to interpolate all the section data onto a regular grid from 0–350 m and 19.9–22.4°N with a grid spacing of 5 m vertically and 0.02°≈2.2 km horizontally.

## 2.2 Vessel mounted sensors

The thermosalinograph data (Strass and Rohardt, 2019) from the water intake of RV *Polarstern* at 11 m depth was processed with standard methods as described in Rohardt (2018).

Furthermore, we used ocean velocity data from a RDI Ocean Surveyor 150 kHz ADCP (VMADCP) mounted in the hull of

RV *Polarstern* 11 m below the surface (repository of the raw data: Witte, 2019). Using the Ocean Surveyor Sputum Interpreter (OSSI) software developed by GEOMAR, it was processed to 2 min averages between 19 m and 251 m depth (for an example of the use of an identically processed data set, see von Appen et al., 2018). Mean volume backscattering strength (MVBS) was calculated from the echo intensity of the VMADCP using the method of Deines (1999).

Surface currents were measured with an X-band marine radar mounted on RV *Polarstern*. The data acquisition and process-

ing was done using the WaMoS 2 system as described in Hessner et al. (2019). The ADCP velocity (and MVBS) gridding was only done from 20–250 m. Velocities in the grid closer to the surface than 20 m were added like this: 0–10 m gridded velocities are the WaMoS velocities, 10–15 m are 2/3·WaMoS velocities + 1/3·ADCP velocities at 20 m, and 15–20 m are 1/3·WaMoS velocities + 2/3·ADCP velocities at 20 m.

## 2.3 Water sample data

Water samples were taken at six depths at the CTD stations prior to and after the Triaxus transect and continuously every 3 h (when the ship was moving) from the ship's seawater system (teflon tubing with a membrane pump). The CTD sampling depths varied, but were chosen to represent the vertical variation of the phytoplankton distribution, containing at least one sample below the chlorophyll-*a* maximum.

The samples for dissolved inorganic nutrient concentrations including nitrate, phosphate, and silicate were filtered through

a 0.2 $\mu$m syringe filter and stored at -80°C while at sea. Nutrient concentrations were assayed on an Evolution 3 Alliance Autoanalyser and calibrations were corrected for concentrations using certified reference material (CRM) 7602a (JAMSTEC; Japan) and MERCK STD (NIST-Standard). Concentrations were calculated by means of standard colorimetric techniques (Grasshoff et al., 2009). At each sampling point duplicates of 4 L of seawater were filtered through a pre-combusted 0.7 $\mu$m 25 mm GF/F filter in order to assess particulate organic carbon (POC) (and particulate nitrogen (PN)). Samples were snap

frozen in liquid nitrogen and stored at -80°C while at sea. Total carbon and nitrogen elemental analyses were performed on an EuroEA Elemental Analyzer (Eurovector EA 3000; Italy).

For phytoplankton pigment analysis, immediately after sampling the water samples were filtered through Whatman GF/F filters. The filters were then shocked in liquid nitrogen and stored at -80°C. The soluble organic phytoplankton pigment concentrations were determined in the home laboratory using high-performance liquid chromatography (HPLC) according to the

method of Barlow et al. (1997) adjusted to our temperature-controlled instruments as detailed in Taylor et al. (2011). We determined the list of pigments shown in Table 2 of Taylor et al. (2011) and applied the method by Aiken et al. (2009).

The chlorophyll-*a* concentrations of the same seven major phytoplankton groups as from the radiometry were calculated based on diagnostic pigment (DP) analysis developed by Vidussi et al. (2001) where we followed Losa et al. (2017) and then modified that according to Bracher et al. (in review) to also derive the chlorophyll-*a* for *Prochlorococcus*. The chlorophyll-*a*

of *Prochlorococcus* sp. was directly given by the divinyl chlorophyll-*a* concentration. The total chlorophyll-*a* was determined from the sum of monovinyl- and divinyl-chlorophyll-*a* and chlorophyllide-*a* concentrations.

## 2.4 Satellite data

The delayed time product merged from all satellites called "Global Ocean Gridded SSALTO/DUACS Sea Surface Height L4" and derived variables were downloaded from http://marine.copernicus.eu. We used the 1/4° gridded sea level anomaly (SLA) product which provides daily values where the data for each day was calculated from the along track data within ±3.5 days of the nominal date. Hence, the gridded values for consecutive days are based on partially overlapping data and are consequently not independent. We also used surface geostrophic currents calculated from the absolute dynamic topography (SLA + geoid).

The NOAA Daily Optimum Interpolation Sea Surface Temperature (Reynolds et al., 2007) was downloaded from https://www.esrl.noaa.gov/psd/data/gridded/data.noaa.oisst.v2.highres.html. We used the 1/4° daily optimally interpolated values merged from available infrared (cloud affected) and microwave (cloud penetrating) satellite data.

The Ocean Colour Climate Change Initiative (OC-CCI) chlorophyll-$a$ dataset, Version 3.1, of the European Space Agency (Sathyendranath et al., 2018) was downloaded from https://esa-oceancolour-cci.org. These are 8-day composites of merged sensor (MERIS, MODIS Aqua, SeaWiFS LAC and GAC, VIIRS) products.

## 3 Results and discussion

### 3.1 Remote sensing information

The sea surface temperature (SST) during the Triaxus section is shown in Figure 2a with the cruise track around 21–22°W. Cold upwelled water was present near the coast. There was also some water with colder temperatures around 21–21.5°N advected out from the upwelling region to the west roughly along the westward advection pathway indicated by the thick magenta line in the figure. The underway thermosalinograph data reveals similar temperatures, but the gradients were sharper than what can be resolved with the gridded satellite data (compare also to Figure 4a below).

The sea level anomaly (SLA) (Figure 2b) reveals the presence of an anti-cyclonic eddy centered on ≈20.5°W 21.5°N. Anti-cyclonic geostrophic flow encircled the sea level maximum. The shipboard measurements along the cruise track from both the VMADCP and the WaMoS radar (green/magenta in Figure 2b) agree well with the geostrophic flow from the satellites except for a deviation of the WaMoS velocity at the northern edge of the eddy. (We refer to Hessner et al. (2019) for a more complete assessment of the differences between WaMoS and VMADCP data; they document a good qualitative and quantitative agreement for all data from the same cruise as discussed here.) The thick magenta line highlights a westward advection pathway that extended from close to the coast to the cruise track between the two anti-cyclones (local SLA maxima) and the elongated cyclonic tongue to their south (depressed SLA). The water within that advection pathway was colder than the ambient water at the same longitude to the north or south. However, surface fluxes (such as solar warming) and possible lateral entrainment during the westward advection likely increased the temperature of the water along the pathway from much colder temperatures in the upwelling region close to the coast. Depending on the dynamics and water mass anomalies, this advection pathway could also be called a "mesoscale filament".

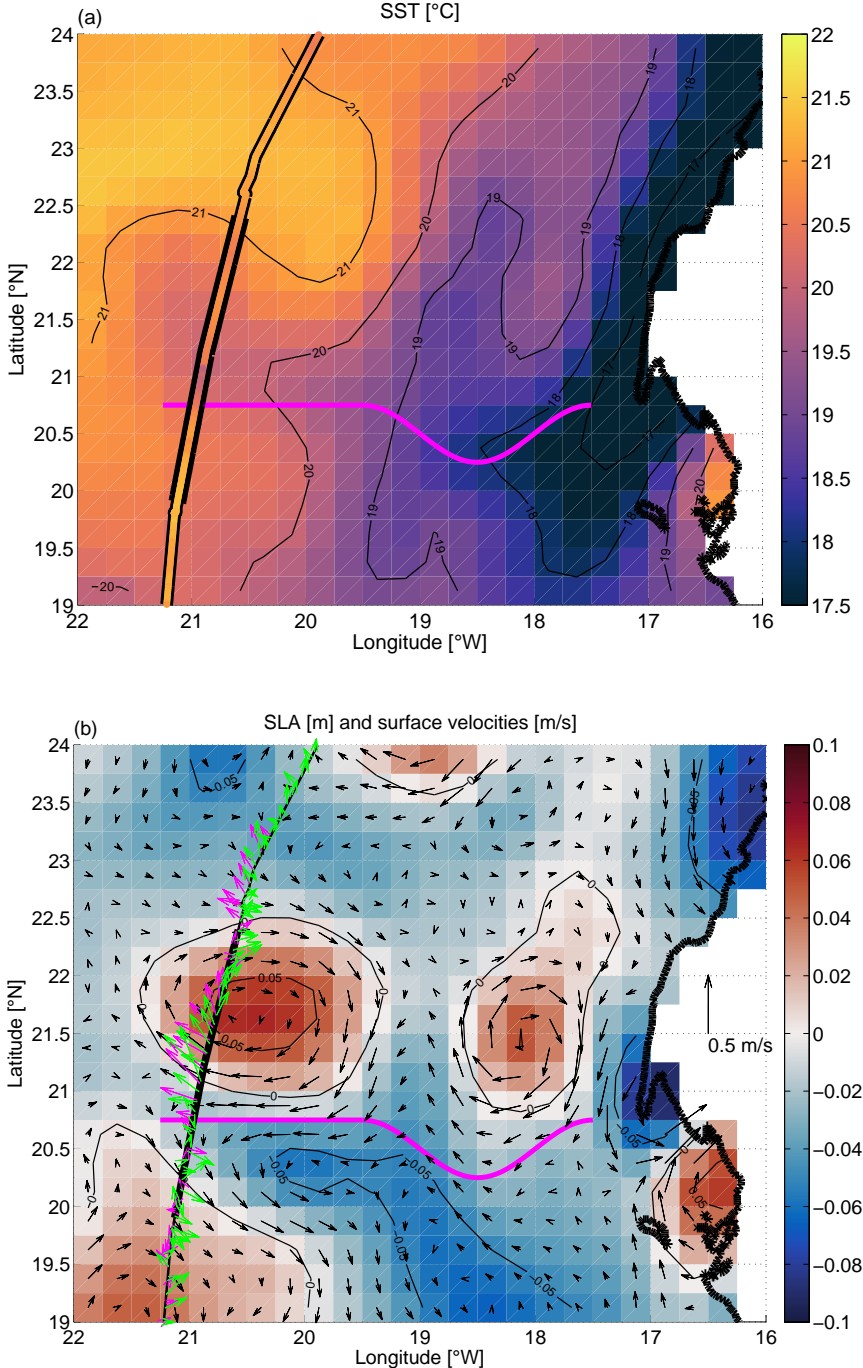

**Figure 2.** Maps of satellite based (a) sea surface temperature [°C; NOAA OI SST v2] and (b) sea level anomaly [m; SSALTO/DUACS SSH L4]. Both SST and SLA are nominally from 31-May-2018, i.e. during the Triaxus tow. The African coast is plotted black around 17°W. The ship track around 21°W is shown in black and in (a) data from the thermosalinograph is plotted between the black lines. The thick part of the track lines shows the towed section presented here. (b) also shows geostrophic velocities [m/s] in black with a 0.5 m/s scale bar shown on the right over land. Green/magenta arrows plot the VMADCP/WaMoS measured velocities in 51 m and 0 m depth respectively. The thick magenta line in both figures approximates the westward advection pathway discussed in the text.

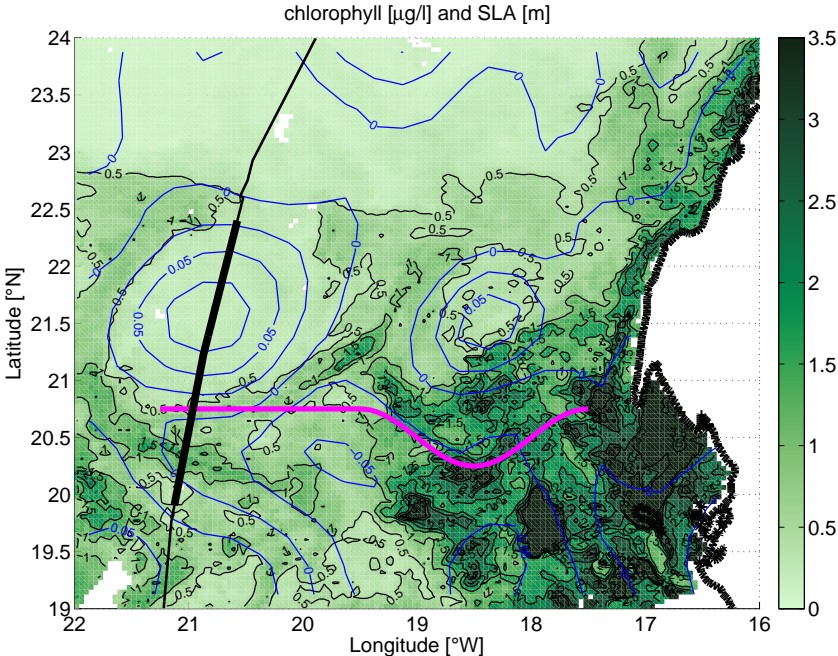

**Figure 3.** Map of satellite based chlorophyll concentration [$\mu$/l; OC-CCI v3.1 product] as color with sea level anomaly [m; SSALTO/DUACS SSH L4] as blue contours. The chlorophyll is an 8-day average centered on 13-Jun-2018 12:00 UTC and the SLA is a 7-day average centered on 13-Jun-2018 00:00 UTC, i.e. approximately two weeks after the Triaxus tow. The ship track and the westward advection pathway are plotted as in Figure 2.

Tracking of the positive sea level anomaly of the anti-cyclone back in time (see supplementary material) revealed that, roughly one month prior, the anomaly had merged from two distinct anomalies. One of those anomalies appears to have formed at 20.5°W 21.5°N two months prior to our section while the other clearly originated in the upwelling region 2.5 months prior to our section. The distance of ≈360 km that it traversed in 2.5 months corresponds to a translational velocity of ≈0.06 m/s. The peak azimuthal velocities of the anti-cyclone are ≈0.5 m/s indicating that the translational velocity is much smaller than the azimuthal velocity. Based on this we speculate that the anti-cyclone had a kinematically trapped core which was able to transport the water in the core for great distances. During this transport the isolated core was subjected to biogeochemical processes (e.g. Karstensen et al., 2017).

High resolution SST and chlorophyll concentration (from ocean color) data could provide a much better synoptic view of the (sub-)mesoscale features discussed in this paper. Those sensors are affected by cloud cover and therefore no such high resolution data is available in the study area within a few days of our 22.5 hour long transect. However, the average from 10-Jun-2018 to 17-Jun-2018 (Figure 3) which is approximately two weeks after the Triaxus transect contains sufficient information to be interpretable. The recently upwelled cold water near the coast is high in chlorophyll. The mesoscale filament providing the

westward advection pathway is also associated with elevated chlorophyll concentrations. At the longitude of the transect, it has moved slightly southwards from approximately 20.9°N (e.g. Figures 4b/7d discussed below) to 20.4°N within the two weeks between the transect and the available satellite derived chlorophyll map (Figure 3). It is also clear that the two anti-cyclones (closed blue contours) are associated with very low chlorophyll concentrations.

We now turn to the section data revealing the results of these processes in the filament and eddy.

## 3.2  Physical properties

The Triaxus section from 19.9°N to 22.4°N and from the surface to 350 m is shown in Figure 4. Potential temperature (Figure 4a) and salinity (Figure 4b) reveal spatial structures and small scale aspects that were resolved at much higher resolution than would have been possible with traditional CTD casts or with satellite observations. These spatial structures include an
eddy, filaments, and mixed layer instabilities which we will present and discuss in detail below. In particular there was slightly colder water at the surface near 21°N which was also much fresher than the water elsewhere at the surface. That was the mesoscale filament of upwelled water which moved westward as seen (and highlighted) in the remote sensing information. The anti-cyclone was located to the north of the filament as will become clear below from the other parameters.

Potential density (Figure 4c) and isopycnals (magenta lines in all section plots) reveal downward sloping isopycnals at the
lower side (below ≈150 m) of the anti-cyclone (between 21.1°N and 21.9°N) and upward sloping isopycnals on its upper side (above ≈150 m). The stratification (Figure 4d) in the center of the anti-cyclone (21.5°N, 200 m) was slightly less than in the rest of the section at that depth marking the mode water (region of relatively weak stratification with upward displaced isopycnals above and downward displaced isopycnals below) that the eddy was focussed around. The stratification also highlights the mixed layer depth (cyan in Figures 4c/d) which was determined from a density difference with respect to the surface of
0.05 kg/m$^3$. The mixed layer depth ranged from roughly 20 m to 70 m, i.e. from quite shallow around 20.5°N to much deeper on the rim of the anti-cyclone (21.2°N and 21.9°N). The density in the mixed layer within the anti-cyclone was significantly higher than waters further to the south. This implies the presence of a significant horizontal density gradient in the mixed layer. These sections also show that the mesoscale filament near 21°N corresponded to sloping isopyncals between upward doming in the south and downward displacements to the north.

A temperature-salinity plot of all measurements of the section (Figure 5a) highlights the isopycnal mixing between North Atlantic Central Water (NACW) which is warm and salty reminiscent of the subtropical North Atlantic, and the South Atlantic Central Water (SACW) which is fresher with its properties probably influenced by the tropical Atlantic or by AAIW.

An isopycnal decomposition of the measured temperature and salinity (Figure 4a/b) into SACW and NACW (as defined by the lines in Figure 5a) shows where these water masses were located (Figure 5b). Around 20.4°N the slanted front between
SACW to the south and NACW to the north becomes clear, but this front had no dynamic signature given that the isopycnals were roughly flat. The anti-cyclone north of 21°N was comprised of fairly pure NACW with a concentration maximum in its mode. The strong isopycnal mixing of the Cape Verde Frontal Zone (the confluent region of NACW and SACW: Tomczak, 1981; Martínez-Marrero et al., 2008) is evident in the large horizontal gradients over a few kilometers between 20.4° and 21.0°N where advected blobs of water often only occupy ≈10 km and a part of the upper 350 m. SACW was most pronounced

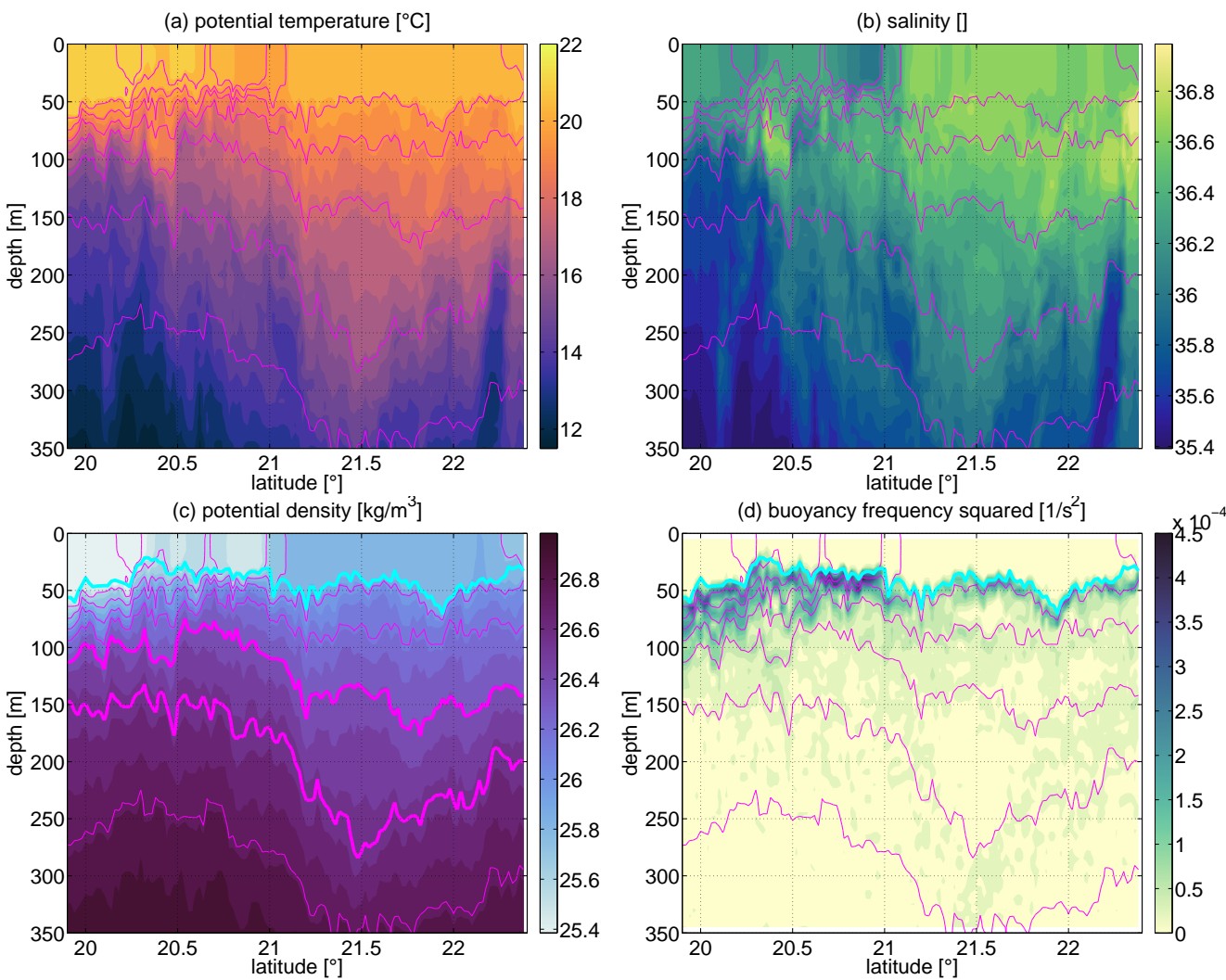

**Figure 4.** Sections of (a) potential temperature [°C], (b) salinity [], (c) potential density [kg/m³], and (d) stratification (buoyancy frequency squared) [1/s²]. In this and all the following figures, the magenta lines are isopycnals plotted at a spacing of 0.2 kg/m³; the 26.4 and 26.6 kg/m³ isopycnals are plotted as thick lines in (c). The cyan line here and below indicates the base of the mixed layer as determined from a density difference with respect to the surface of 0.05 kg/m³.

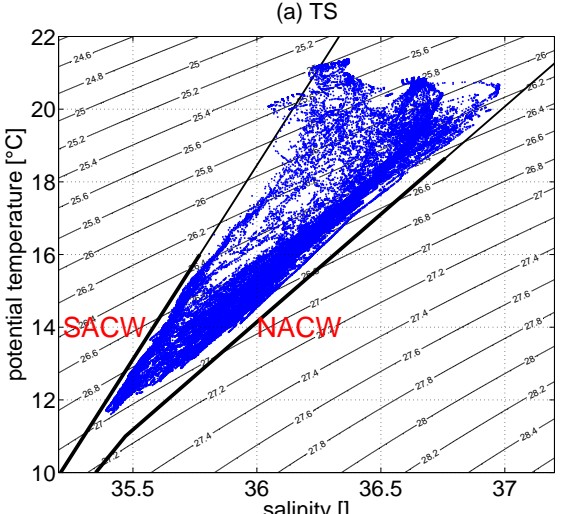 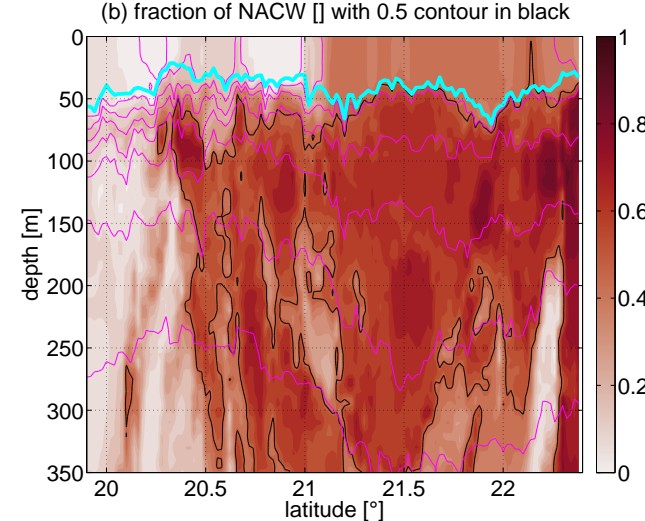

**Figure 5.** (a) Temperature-salinity diagram of all measured CTD data with the fresher South Atlantic Central Water (SACW) and the saltier North Atlantic Central Water (NACW) end member lines according to Tomczak (1981) marked in black. (b) Section of warm-salty North Atlantic Central Water (NACW) end member fraction for an isopycnal decomposition between SACW and NACW. The 0.5 contour is indicated in black. The mixed layer depth and isopycnals are as in Figure 4.

in the mixed layer in the southern part of the section, but it also contributed roughly half to the mixed layer in the northern part of the section. A narrow band (<20 km) of SACW at depth was also present north of the anti-cyclone (≈22.2°N).

Next we will examine the dynamics starting with the velocity structure. We investigate the velocity in the direction of the cruise transect ("along-track velocity", Figure 6a) where positive values roughly correspond to northward velocity as well as the velocity perpendicular to the direction of the cruise transect ("cross-track velocity", Figure 6b) where positive values

roughly correspond to eastward velocity. The ship track clearly passed through the western side of the anti-cyclone. Northward flow throughout most of the feature (21–22°N) is observed with westward (negative cross-track) flow on its southern side (south of 21.5°N) and eastward flow on its northern side. This mesoscale (order of 50 km) coherent flow structure dominated the northern part of the section. In contrast, south of 21°N, there was a lot of variability in the velocity on smaller scales of 1–5 km.

Since we do not resolve the velocity structure in the cross-track direction, we can only estimate one component of the relative vorticity $\zeta = \frac{\partial v}{\partial x} - \frac{\partial u}{\partial y} \approx -\frac{\partial u_r}{\partial y_r}$, where $u_r$ is the cross-track velocity and $y_r$ is the along-track coordinate. This estimate of the relative vorticity normalized by the Coriolis frequency $f$, i.e. the Rossby number $Ro$, (Figure 6c) shows strong velocity gradients at the base of the mixed layer in the southern part of the section with relative vorticities of up to $\pm 1$. Since mesoscale flows would only have $|Ro| \ll 1$ and the large $Ro$ clearly appear to be located at the base of the mixed layer, this suggests that

submesoscale mixed layer eddies or filaments were present there associated with distinct extrema of the NACW/SACW water mass fraction. The radius of these is 5–10 km, which is much less than the first baroclinic Rossby radius ($\frac{N}{f}H$) of ≈45 km

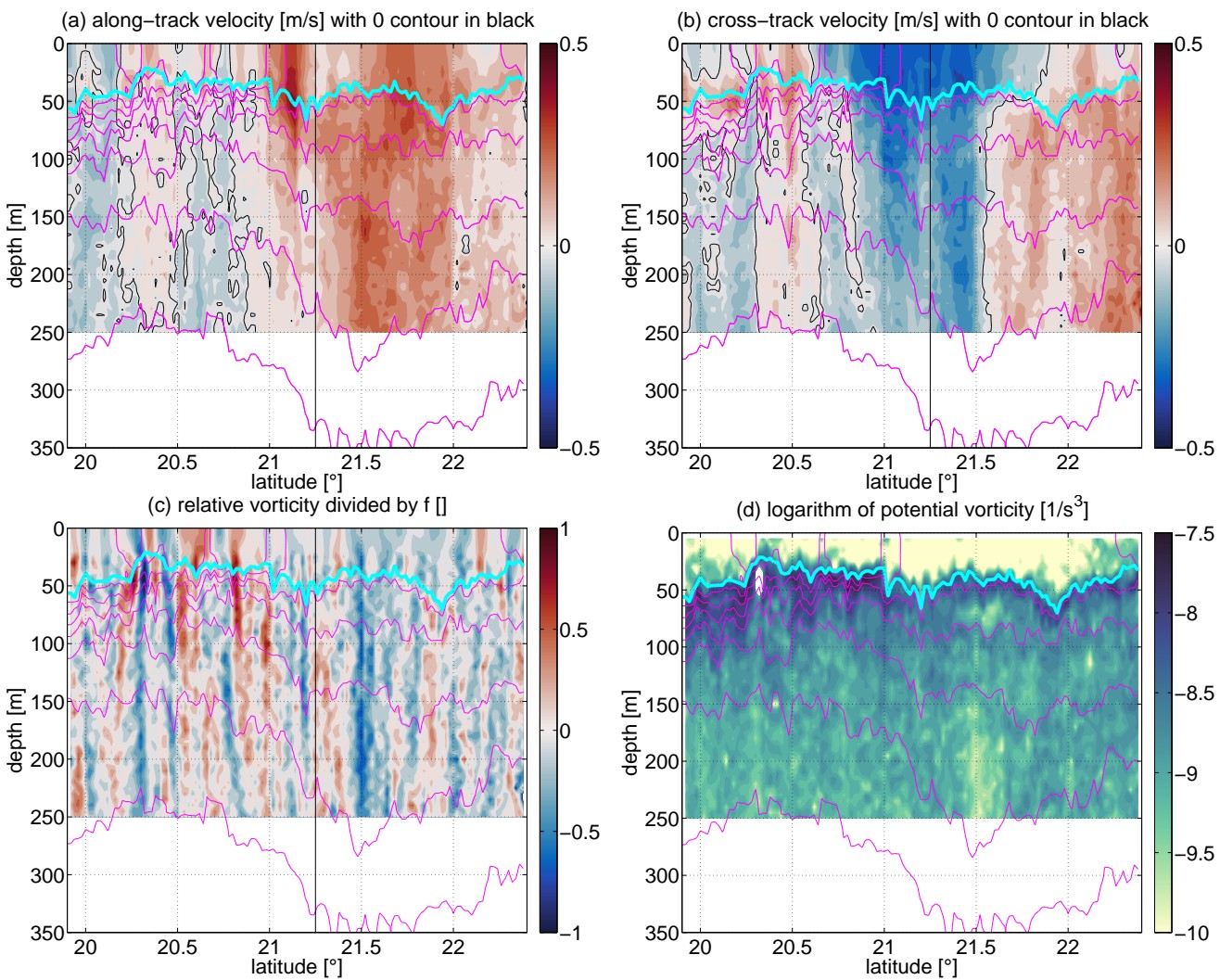

**Figure 6.** Sections of (a) along-track velocity [m/s] (positive roughly northward), (b) cross-track velocity [m/s] (positive roughly eastward), (c) Rossby number (relative vorticity divided by f) [], (d) logarithm of potential vorticity [1/s³]. The vertical black line at 21.25°N highlights where the course of the section turned by 4° from 10.5° to 14.5° with respect to true north.

in the region (Chelton et al., 1998) and closer to the mixed layer Rossby radius ($\frac{N_{ML}}{f} H_{ML}$), where $H$ and $N$ are the full water column depth and the stratification averaged over the full depth and $H_{ML}$ and $N_{ML}$ are those quantities averaged over the mixed layer. The potential vorticity $PV = N^2 * (f + \zeta)$ (plotted logarithmically in Figure 6d) shows the effect of wind destruction of PV in the mixed layer. It also picks out the near zero PV core of the anti-cyclone at 21.5°N, 200 m. Since PV in the ocean below the mixed layer is conserved except for mixing or other diapycnal processes, it seems that the water in the core came from a location where its PV was forced to zero, i.e. likely under wind or negative buoyancy (cooling) forcing. In other words, we can use PV as a passive tracer of a water parcel's history. This is another piece of evidence suggesting that the mode water in the ACME originated in the upwelling region. At 20.3°N at 50 m (the base of the mixed layer), PV was negative (it appears white in the figure as it is outside of the range of the positive logarithmic plot). This is an indication for symmetric instability which rapidly overturns and thereby brings the system back to neutral levels of zero PV (marginal stability). In the process it likely induced vertical mixing with implications for biogeochemistry (discussed in more detail in Subsection 3.5 below). At 20.4°N it appears to also have recently injected fluid into the mixed layer as PV≠0 there (if fluid would have stayed in the mixed layer for longer, the wind would have driven its PV to zero).

## 3.3 Biogeochemical properties

Now we turn to the observed biogeochemical structure. Oxygen concentrations (section in Figure 7a; averaged over the top 20 m in Figure 8a) and nitrate concentrations (Figure 7b/8a) were measured by two independent sensors with different measurement principles. Phosphate and silicate are not thought to be the limiting nutrients in this region (Bachmann et al., 2018) as is also suggested from the high phosphate and silicate concentrations detected in the underway (Figure 8a) and CTD-rosette samples. Thus we consider nitrate as sufficient to study the nutrient availability for phytoplankton growth. In order to highlight similarities, the color axes between Figures 7a/b are reversed as nitrate is used up and oxygen is released during primary production. Thus as long as biology remains the primary influence on the two parameters, their relation follows a roughly constant ratio which is typical for much of the world ocean and is related to the equivalent of a Redfield ratio (Thomas, 2002; Waite et al., 2015) and reflects the biogeochemical uptake of nitrate and stoichiometric release of oxygen during photosynthesis. On first sight there is a lot of agreement between oxygen and nitrate overlaid on a pronounced variability (Figures 7a/b). Following individual isopycnals below the mixed layer, SACW was more nitrate rich/oxygen poor than NACW suggesting that SACW may be older (having been removed from atmospheric fluxes in the mixed layer for a longer duration). The red lines in Figure 7a indicate 100% oxygen saturation. Apparent oxygen utilization (AOU) is the oxygen concentration at 100% oxygen saturation (i.e. in diffusive balance with the atmosphere) as calculated from the measured temperature/salinity minus the measured oxygen concentration. The mixed layer was super saturated (negative AOU) in places both to the north and south of the anti-cyclone (Figure 8a), suggesting that primary production had supplied oxygen to the mixed layer more recently than the gas equilibration time scale. The nitrate concentration in the mixed layer in this spring/summer condition was very low (lowest underway measurement of 0.01 $\mu$mol/l at 21.6°N), but e.g. at 20.9°N there was actually still a significant amount (up to 5 $\mu$mol/l) in the mixed layer. At depth by contrast, just below 100 m especially in the southern part very nitrate rich values

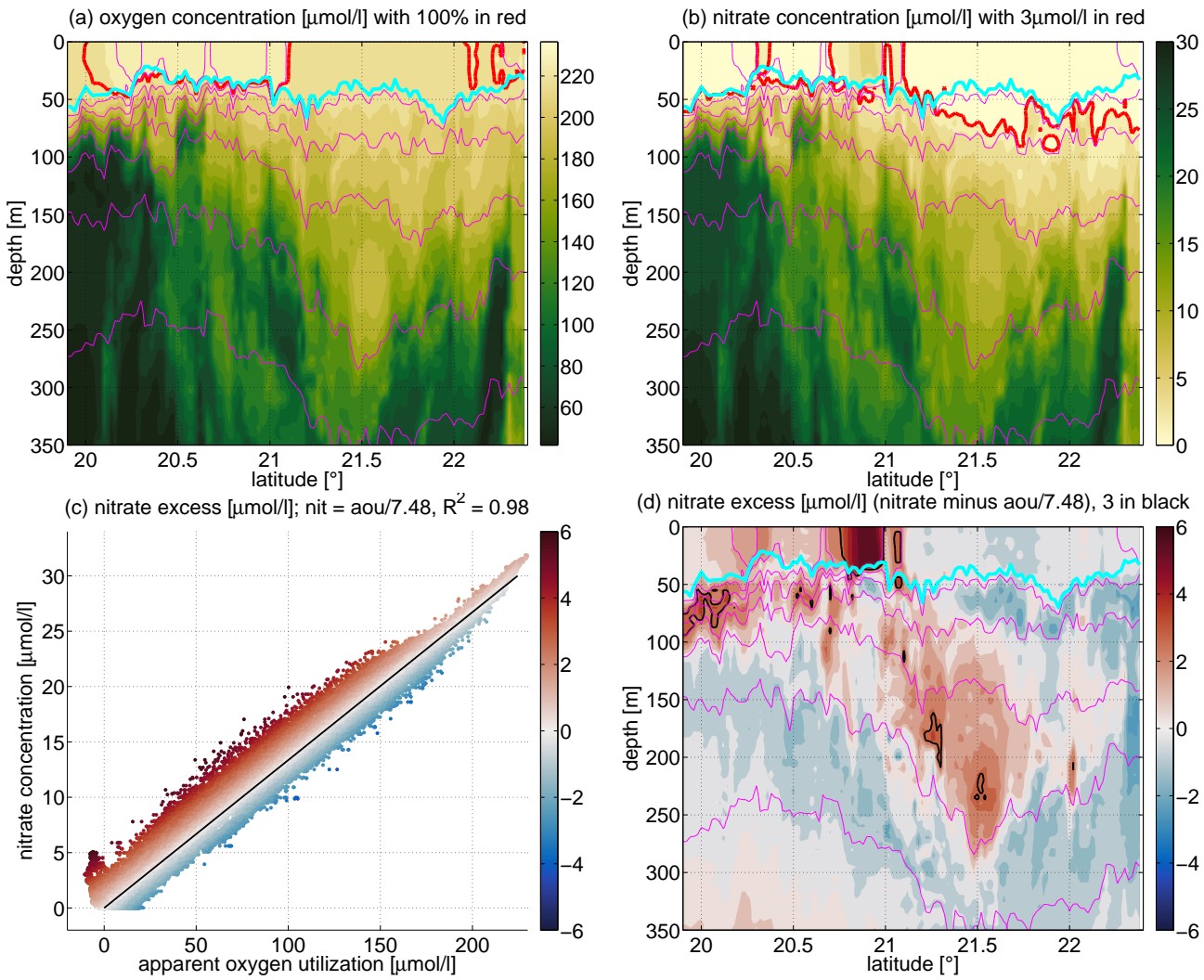

**Figure 7.** Sections of (a) oxygen concentration [$\mu$mol/l], (b) nitrate concentration [$\mu$mol/l], and (d) nitrate excess [$\mu$mol/l]. The following contours are highlighted: (a) 100% oxygen in red, (b) 3$\mu$mol/l nitrate in red, and (d) 3$\mu$mol/l nitrate excess in black. (c) all data in scatter plot of apparent oxygen utilization versus nitrate concentration where the black line is a best fit linear regression nitrate = AOU/7.48. A nitrate excess of zero is on the black line in (c); positive/negative nitrate excess is colored red/blue above/below the black line in (c).

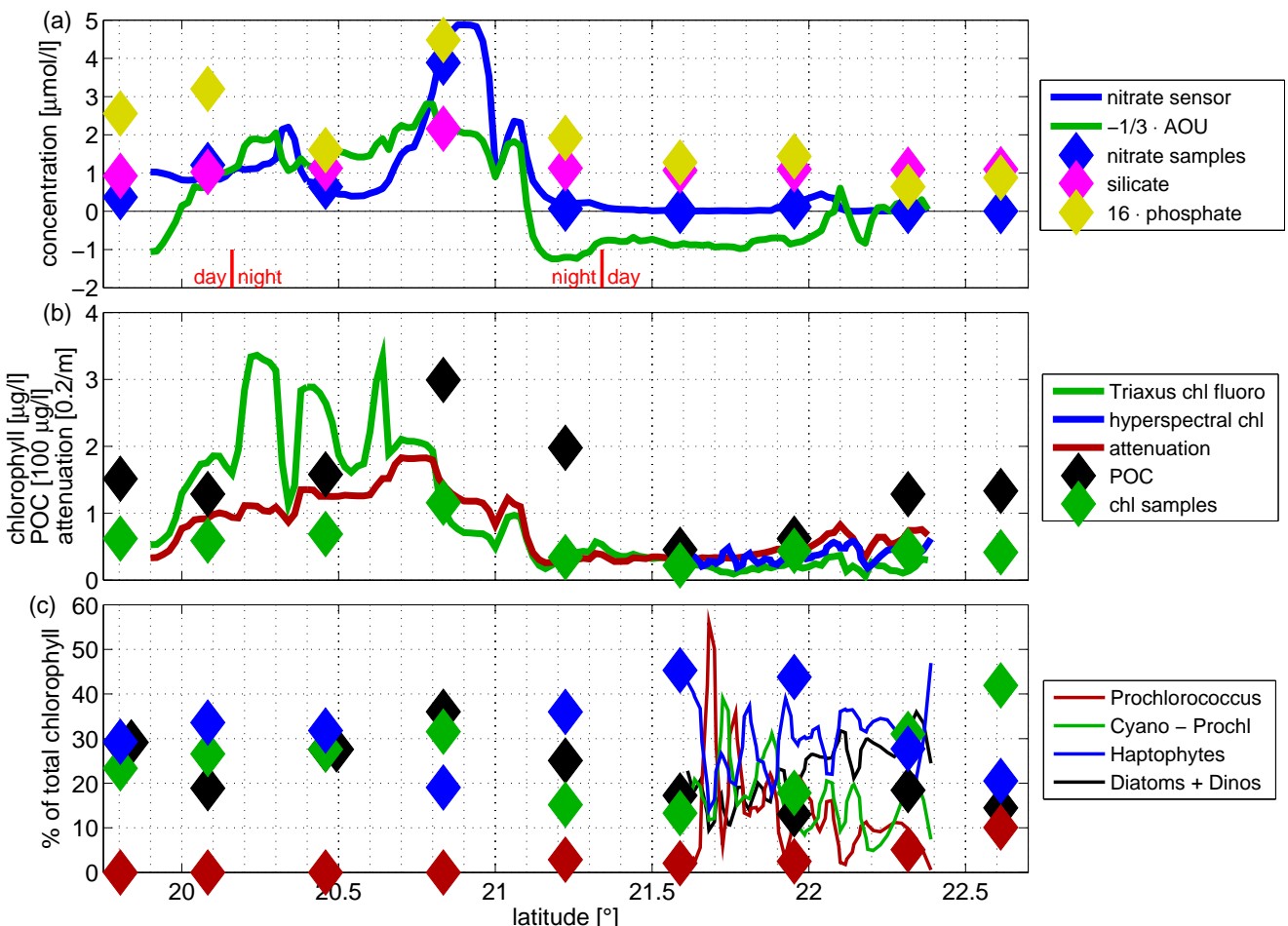

**Figure 8.** Mixed layer properties with sensor measurements averaged over 0–20 m in (a)/(b) and 0–30 m in (c) as lines and samples from 11 m as diamonds. (a) concentrations [$\mu$mol/l] of nitrate (blue), $-1/3 \cdot$ AOU (green), silicate (magenta), and $16 \cdot$ phosphate. (b) chlorophyll-*a* concentration [$\mu$g/l] from the fluorescence sensor on the Triaxus CTD (green line), samples (green diamonds), and the hyperspectral irradiance sensor (blue), attenuation [0.2/m] (red), and particulate organic carbon [100 $\mu$g/l] (black). Note the different units used on the y-axis. (c) percentage of total chlorophyll-*a* accounted for by different phytoplankton functional types: *Prochlorococcus* (red), cyanobacteria excluding *Prochlorococcus* (green), haptophytes (blue), and sum of diatoms and autotrophic dinoflagellates (black). Lines are from the hyperspectral irradiance sensor and diamonds are from HPLC measurements of underway samples.

were reached (>20 $\mu$mol/l, Figure 7b). Such concentrations, if upwelled into the euphotic zone, would have the capacity to support increased productivity.

AOU is plotted versus nitrate concentration in Figure 7c. Most of the individual measurements (colored dots) are close to proportional following the relation nitrate concentration = AOU/7.48. This value is within typical observations of scatter around the Redfield ratio for organic matter (Thomas, 2002; Waite et al., 2015). However, not all individual measurements are directly proportional and the color indicates where there is nitrate excess, e.g. for the dark red dots in the bottom left of Figure 7c; at zero AOU, zero nitrate concentration would be expected but in fact $\approx$5 $\mu$mol/l (i.e. an excess of 5 $\mu$mol/l) was observed. We note that this definition of nitrate excess is only based upon our measurements (Figure 7c).

The spatial distribution of the nitrate excess (Figure 7d) highlights the cool fresh mesoscale filament that came from the east which still had non-zero nitrate values in the mixed layer. A significant nitrate excess is also visible in the core of the anti-cyclone and in blobs in the thermocline in the south. We speculate that positive nitrate excess can originate from mixed layer water that was not equilibrated (equilibration here is understood as being where gas equilibration with the atmosphere has brought oxygen to 100% saturation and primary production consumed the nitrate to zero). Large undersaturation of oxygen (as is common in the upwelled water in the Mauretanian upwelling system) may have led to strong oxygen fluxes (equivalent to a reduction in AOU) from the atmosphere to the mixed layer. At the same time, the residence time in the upper ocean probably was not long enough for primary production to utilize available nitrate before the water was subducted. We hypothesize that the water in the core of the anti-cyclone during our section spent only a short time ($\sim$weeks) in the mixed layer before being subducted (i.e. removed from equilibrating processes in the mixed layer) to form the anti-cyclone. Assuming that the correlation in Figure 7c corresponds to the near Redfield behavior of phytoplankton growth and remineralization in the study area, primary production and remineralization acting in a parcel of water can move it parallel to the correlation line (black line in Figure 7c), but not change its nitrate excess (i.e. move it further away from the line). A change in the nitrate excess can only be achieved by a process that acts differently upon nitrate than upon oxygen. Gas exchange with the atmosphere likely plays that role as it only acts upon oxygen, but not upon nitrate. The gas exchange rate also depends upon how far from equilibrium with the atmosphere (100% oxygen concentration) the parcel of water is. That is, the gas exchange rate will be faster upon upwelling of the extremely undersaturated water thereby pumping oxygen into the water from the atmosphere. This corresponds to a reduction of AOU and a corresponding increase in the nitrate excess. After primary production has acted, the water may be close to saturation or slightly super-saturated. However, because the water is closer to equilibrium then, the loss of oxygen to the atmosphere will be comparatively slower. Thus a nitrate excess can be present in the mixed layer relatively soon after upwelling (as measured by the relatively slow gas exchange near equilibrium). If the water gets subducted (stopping the gas exchange) during that time, a nitrate excess can also be observed later on. Part of this hypothesis is that we assume that a nitrate excess of roughly zero is present in the water when it is upwelled. This is based upon the assumption that, if, in our section, we sample water that will upwell in the future, it would be "old" (high AOU, high nitrate concentration) and would thus fall in the top right quadrant of Figure 7c, where we observe a very small nitrate excess as defined here. In a similar sense, the filament south of 21°N must have been part of an offshore advection "highway" from the upwelling region whose advective time was short enough for the above argument to hold.

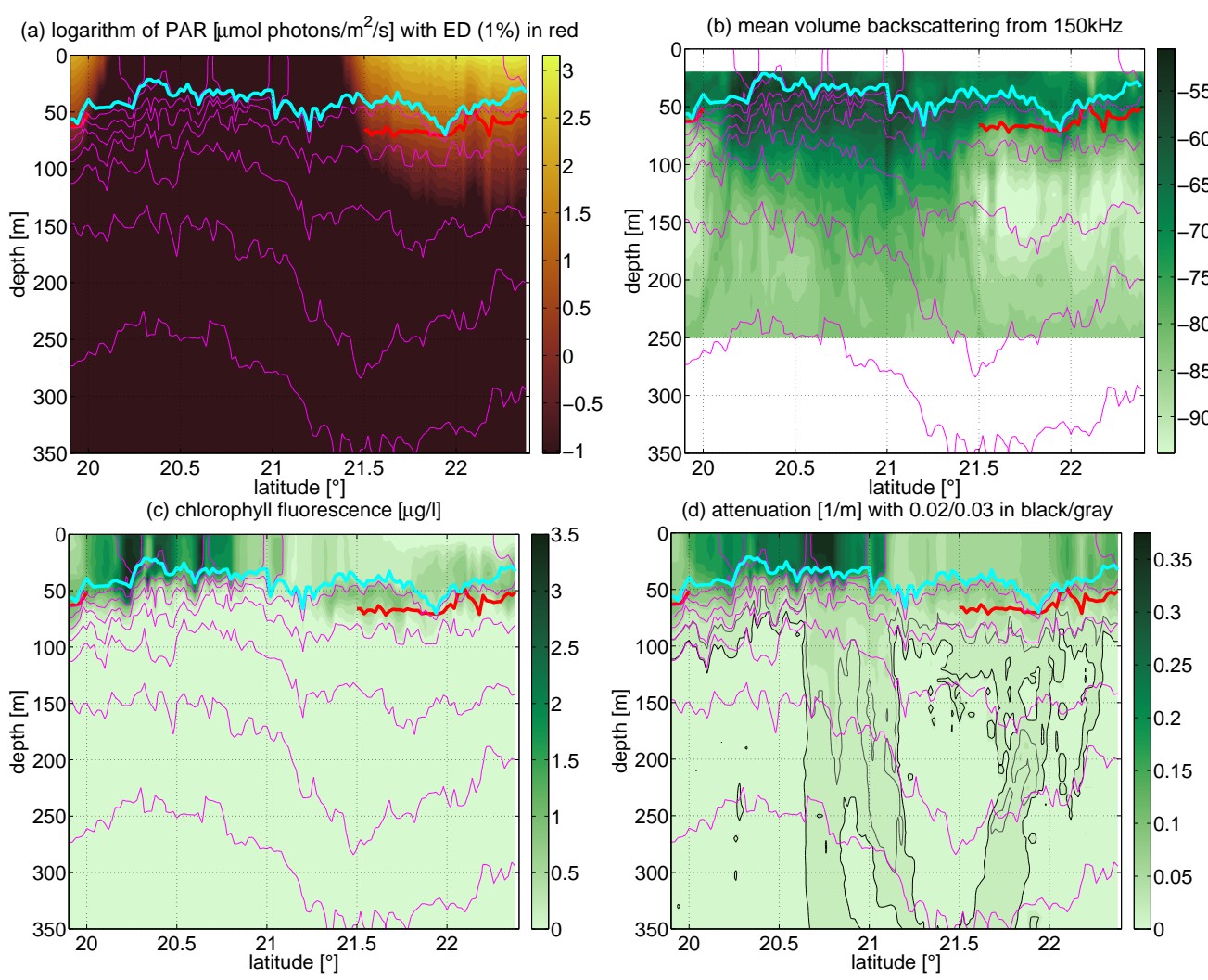

**Figure 9.** Sections of (a) logarithm of photosynthetically available radiation [$\mu$mol photons/m²/s], (b) mean volume backscattering from 150 kHz vessel mounted ADCP [arbitrary units], (c) chlorophyll-*a* fluorescence from the CTD sensor [$\mu$g/l], and (d) light attenuation [1/m]. The red line indicates the euphotic depth (1% of surface PAR). In (d) the 0.02/m and 0.03/m contours are indicated in black and gray respectively.

365     Considering the energy source for primary production, photosynthetically available radiation (PAR) as shown logarithmically in Figure 9a is affected by the sunset around 20.1°N and the sunrise around 21.4°N. During the day time, the euphotic depth (1% level of surface PAR) as shown by the red line varied between 50 m and 70 m. In the southernmost part of the section this limit for primary production coincided with the base of the mixed layer (cyan line). By contrast, in the northern part, light reached into the stratified water column below the nutrient depleted mixed layer.

## 3.4 Biological properties

Phytoplankton standing stock in the water column can be inferred from chlorophyll-*a* fluorescence (Figure 9c) or light attenuation (Figure 9d) with a number of assumptions. Here we compare the chlorophyll-*a* fluorescence sensor that is part of the Triaxus CTD with the underway samples of chlorophyll-*a* concentration from the mixed layer (Figure 8b). The laboratory culture-based calibration of the fluorescence may be off by a factor of 2 due to species specific variations and environmental adaptations and, during day time, photochemical quenching in the upper part of the water column can diminish significantly the chlorophyll-*a* fluorescence signal (Falkowski and Kolber, 1995). In that situation the measured fluorescence is lower than the chlorophyll-*a* concentration present in the water column. This is apparent in the upper 20 m north of 21.7°N. Light attenuation by contrast responds to any matter in the water, but in the open ocean observed here, that is primarily phytoplankton and its degraded particulate organic matter. The chlorophyll-*a* fluorescence from the Triaxus CTD may however be used to study relative changes within a region of the transect. Especially south of the anti-cyclone, chlorophyll-*a* fluorescence and attenuation were high in the mixed layer which also corresponded to large particulate organic carbon (POC) values (Figure 8b) together with considerable spatial variability in chlorophyll-*a* fluorescence.

     In the anti-cyclone the Triaxus chlorophyll-*a* fluorescence data from the section (Figure 9c) indicates a deep chlorophyll-*a* maximum at ≈50–70 m in the stratified water column below the nitrate-deplete mixed layer. However, this may well be an artifact of possible quenching of chlorophyll-*a* fluorescence during day light. In the southern part of the section (where measurements were taken during night time and therefore chlorophyll-*a* fluorescence data are more robust in indicating chlorophyll-*a* concentration), by contrast, the values in the mixed layer were much higher. Maxima of chlorophyll-*a* fluorescence and attenuation were observed where isopycnals were vertical in the mixed layer (Figure 9c/d). At those locations, thermocline waters were outcropping and horizontal density gradients in the mixed layer were present, e.g. at 20.3°N (chlorophyll-*a* fluorescence maximum), 20.6°N (attenuation maximum), and 21°N. These mixing processes may be responsible for the high productivity in the southern sector, and will be discussed in more detail below. North of the anti-cyclone, light attenuation in the mixed layer and chlorophyll-*a* fluorescence also point to favorable growth conditions (this is below the depth where during day time photochemical quenching takes place enforcing the quality of these data). The reduction of the euphotic depth from 70 m just to the south to 50 m at 22.1°N correlates with the increase of attenuation and it therefore may be the result of a higher concentration of phytoplankton and associated colored dissolved and particulate organic matter. This suggests that the euphotic depth south of the anti-cyclone (where we have not measured it due to night time) would have been shallower, thereby inhibiting growth below the mixed layer there.

In the mixed layer, oxygen (Figure 8a) was super saturated (negative AOU) in the southern part and nitrate remained available with especially large excess values at 20.9°N. In contrast, in the northern part nitrate was low, but not entirely zero e.g. around 22.1°N. The attenuation correlates very well with total chlorophyll-*a* concentration determined from the continuously measured spectrally resolved radiometric data via the EOF method (only available for the day time part of our transect) and with the chlorophyll-*a* determined from discrete water samples via HPLC analysis. This is because in these so-called optical case-1 waters (see Morel and Prieur, 1977) the light attenuation is determined mostly by the concentration of phytoplankton pigments and related colored dissolved and particulate organic matter. However chlorophyll-*a* determined from the fluorescence sensor that is part of the Triaxus CTD agrees only qualitatively with the other chlorophyll-*a* estimations due to variation in the signal from the above mentioned photochemical and non-photochemical quenching and the phytoplankton species composition.

The relative contribution to the average surface (0–30 m) total chlorophyll-*a* concentration (Figure 8c) based on the continuous radiometric data set available north of 21.6°N reveals the following. South of 22°N, the prokaryotic phytoplankton groups *Prochlorococcus* and all other cyanobacteria typically accounted for between 30% to 50% with in most cases one of them being the largest group followed by haptophytes. In contrast, north of 22°N their contribution became marginal (<5% for *Prochlorococcus*, <15% for other cyanobateria). Conversely, the larger phytoplankton groups of diatoms and dinoflagellates contributed up to 40% in the northern part and and only less than 20% in the southern part. Haptophytes were mostly the dominant or second dominant group throughout the transect and did not significantly change in their contribution between the southern and the northern part. The following may be an explanation for the observed phytoplankton composition along the transect: The upper ocean system was in a post bloom respiring state as suggested by the oxygen undersaturation and very low nitrate concentration which might have favored haptophytes and at very low nitrate concentration in the southern part especially the growth of the prokaroytic phytoplankton. By contrast, north of 22°N outside of the anti-cyclone where nutrient limitation did not appear to have been as severe and also south of the anti-cyclone, diatoms and dinoflagellates made up a much higher proportion (up to 40%) of the total phytoplankton biomass and thereby the contribution of other cyanobacteria and *Prochlorococcus* is significantly lowered. These submesoscale features were not resolved in the water sample data. As we can see from the comparison of the two phytoplankton composition data, the water sample data north of 22°N only catch the low values for diatoms and dinoflagellates and the higher values for cyanobacteria. However, further south, where high-resolution radiometric based phytoplankton composition data are not available, the water sample data provide further valuable information: In the nutrient rich southern part of the transect cyanobacteria were much more prominent accounting for 20–30%. *Prochlorococcus*, on the other hand, were absent there, probably because their competitive advantage is in nutrient limited stable conditions rather than the dynamic environment that we encountered south of 21.1°N. These findings with respect to the phytoplankton group composition in the surface layer compare well to other observations in the Mauretanian upwelling system (e.g. Taylor et al., 2011).

The light attenuation significantly diminishes below the euphotic layer. Attenuation below approximately 100 m (Figure 9d) tends not to be due to living phytoplankton, but to exported carbon either in higher trophic levels or in sinking particles. At 20.7°N there were high attenuation values which imply high particle concentrations roughly oriented in a vertical manner below the high nitrate/chlorophyll-*a* mixed layer. These potentially stemmed from fecal pellets produced by zooplankton feeding on

the blooming phytoplankton. The attenuation further suggests that there were enhanced particle concentrations on the rim of the anti-cyclone, but with increasing depth the concentration maximum appears to be closer to the center of the anti-cyclone. This is reminiscent of the "wine-glass" mechanism proposed by Waite et al. (2016) whereby secondary circulations associated with mesoscale eddies transport particles inwards as they settle in the frame of reference of the eddy.

Acoustic mean volume backscattering at 150 kHz from the vessel mounted ADCP (Figure 9b) is primarily due to zoo-plankton and over a day-night cycle can map its diel migration. Throughout the night backscattering in and below the mixed layer was enhanced suggesting that zooplankton may have been feeding on the phytoplankton there. However, the day-night cycle makes the acoustic mean volume backscattering difficult to interpret (Vélez-Belchí et al., 2002; Cisewski and Strass, 2016). Simply following Figure 9b suggests that the zooplankton concentration at 21.1–21.3°N, i.e. in the low productivity anti-cyclone was lower than in the productive frontal region south of it, but this may also entirely be an artifact of the diel migration.

### 3.5 Submesoscale structure

We now investigate in more detail the physical processes that may have contributed to supporting the higher productivity away from the influence of the anti-cyclone. A spatial and colorscale zoom on the upper water column of 55 km in the south of the section is shown in Figure 10. Around 20.3°N the mixed layer shoaled from 50 m to 20 m over a horizontal distance of under 10 km with isopycnals north of this location steeply dropping. Geostrophy led to the opposing cross-track velocities of $\pm 0.25$ m/s with the cross-track velocity extrema in the stratified water column just below the mixed layer. There the stratification was able to sustain the vertical shear without the system becoming Kelvin-Helmholtz unstable (Richardson number $\lesssim 1/4$). The horizontal velocity shear resulted in the large relative vorticity divided by $f$ of up to $+1$ and below $-1$ with the region of $\zeta/f < -1$ and consequently negative potential vorticity at 20.32°N (Figure 6d). We hypothesize that this made the region susceptible to rapid readjustment via symmetric instability associated with fluid exchange in the vertical and horizontal. Given that also PV (a dynamically active parameter, Figure 6d) was enhanced at the surface, a simple diffusive process bringing e.g. nitrate to the surface is not sufficient to explain the observations. Rather the vertical exchange likely also contained an exchange of fluid between the mixed layer and the thermocline below. From the available measurements it is not possible to distinguish whether this mixed layer/base of the mixed layer feature was an eddy, a front, or a filament, but it is clear that very active dynamics were associated with it.

The observation of negative potential vorticity suggests that the situation we observed is likely to have been ongoing for only 1–2 days. That time scale is similar to the phytoplankton doubling time in that part of the ocean meaning that a large number of doublings and associated POC increase/nutrient draw-down likely did not take place during the time that the situation had been ongoing. The biogeochemical measurements (Figure 10d-f) corroborate this. In the mixed layer less than a few kilometers north of the region of negative potential vorticity (i.e. north of 20.3°N), the nitrate values were enhanced ($\gtrsim 2\mu$mol/l) with lower (night-time) chlorophyll-$a$ values than elsewhere in the mixed layer and slightly lower oxygen saturation. This suggests that the nutrient-rich water from below the euphotic zone had been recently injected into the mixed layer with phytoplankton unable to take the nutrients up in the time available since injection. By contrast, it is likely that the dynamic physical environment had

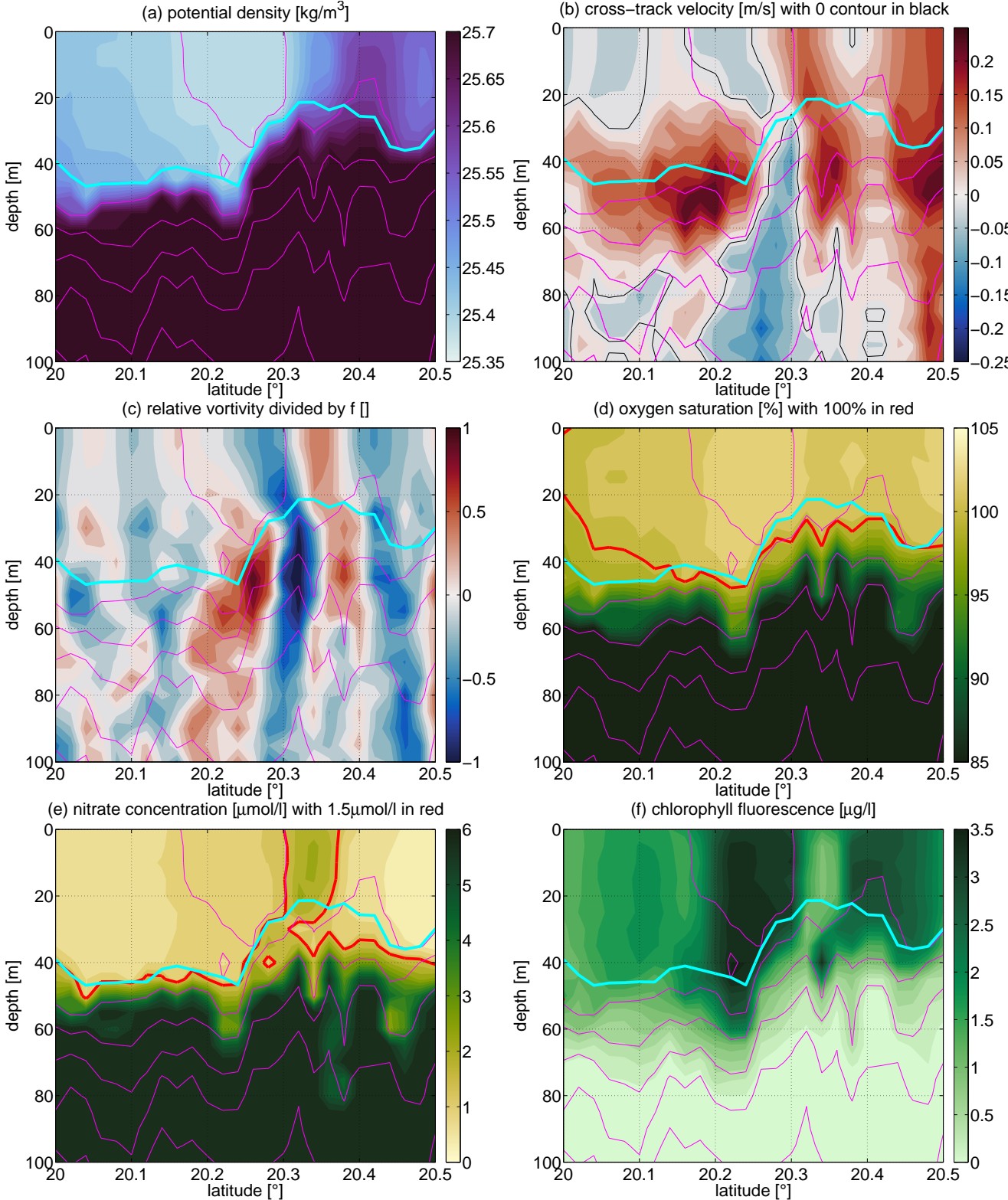

**Figure 10.** Sections zoomed in on the upper 100 m over 55 km in the south of the transect. (a) potential density [kg/m³], (b) cross-track velocity [m/s], (c) relative vorticity divided by f [], (d) oxygen saturation [%], (e) nitrate concentration [μmol/l], and (f) chlorophyll-*a* fluorescence [μg/l].

impacted the mixed layer both 5–10 km to the south and to the north in a similar manner (e.g. symmetric instability), several days to a few weeks earlier. We speculate that phytoplankton there grew to large concentrations which increased oxygen and drew down the nitrate. However, the achieved nitrate values are not nearly as close to zero as above the anti-cyclone where the slower physical evolution of the system over months probably provided phytoplankton of all size classes sufficient time to reduce nitrate to the observed values of as little as 0.01 $\mu$mol/l.

## 4  Conclusions

In our study, we have shown the structure of an $O(100$ km$)$ diameter anti-cyclonic mode water eddy, of an $O(20$ km$)$ width mesoscale upwelling filament, and of several $O(5–10$ km$)$ wide submesoscale mixed layer instabilities in the region offshore of the Northwest African upwelling system. We used the physical observations to suggest explanations for the observed differences in the biogeochemistry and biology along the section both in the mixed layer and below. The single transect without cross-track resolution provides a snapshot of the processes that control the scales of biogeochemical variability in the region, but the lack of cross-track resolution does not allow us to scale this up to quantify the impact of these processes for the region.

The driving mechanism for the mixed layer eddies/filaments in the southern half of the section is not fully clear. The initial creation of negative potential vorticity (PV) is likely to be a key driver. It seems highly probable that once the negative PV was created, symmetric instability would have restored the potential vorticity back to zero. Downfront winds may drive the PV to negative values, but at this point we lack a detailed understanding of the orientation of the fronts with respect to the wind. This would require surveys resolving two horizontal dimensions by a number of parallel sections.

The intricate interplay of different parameters at kilometer scale needs to be taken into account when interpreting single profile and/or bottle data in such dynamically active regions of the ocean. Additionally, our observations reinforce the notion that the coupling and decoupling of biological and physical time scales plays a key role driving spatial variability. We documented physical forcing ranging in spatial scale from kilometers to hundreds of kilometers, covering time scales ranging from days (for mixed layer instabilities) to months (for ACME advection). These variations then interact with plankton communities at physiological time scales (hours to days) and ecological (weeks) time scales, resulting variously in either an increase (where coupled) or a smoothing (where uncoupled) of the spatial variability. Finally, this study has demonstrated that the Triaxus platform in its present setup is an ideal tool for interdisciplinary research, especially where interesting physical and biological/biogeochemical dynamics with spatial gradients exist such as eddies, fronts, and filaments.

*Data availability.* The Triaxus, CTD, VMADCP, and thermosalinograph data are available at the Pangaea references listed in Section 2. The satellite data are available at the URLs listed in Section 2. The following data is provided as Supplementary Material:

**Supplementary material 1: table_suppl_material.xlsx**: Nutrient, POC, total and phytoplankton group chlorophyll-*a* concentration data from underway samples at 11 m and averaged over 0–30 m. Total chlorophyll-*a* is the sum of the seven phytoplankton groups which can be predicted from the RAMSES sensor data.

**Supplementary material 2: table_suppl_material_WaMoS.xlsx**: WaMoS surface current data.

*Video supplement.* **SSH_animated.pdf**: Animation of sea level anomaly and surface geostrophic velocities in the region as in Figure 2b over the 2.5 months preceding the section presented here. pdf-file needs to be opened with Adobe Reader for animation to work.

*Author contributions.* W.-J.v.A. conceived the analysis and wrote the manuscript. V.H.S. conceived the measurements. W.-J.v.A., V.H.S., A.B., H.X., and C.H. collected and analyzed the data. M.H.I. and A.M.W. assisted in the analysis. All authors interpreted the data and commented on the manuscript.

*Competing interests.* The authors declare that they have no conflict of interest.

*Acknowledgements.* We thank H. Becker, S. Spahic, J. Hagemann, and S. El Naggar for their invaluable help in collecting and processing the data presented here. Support for this study was provided by the Helmholtz Infrastructure Initiative FRAM. We thank S. Wiegmann for supporting phytoplankton pigment water sample collection and radiometry data. We thank S. Wiegmann and B. Bracher for HPLC sample analysis. We acknowledge ACRI-ST for supporting H. Xi's participation in PS113 via the project OLCI-PFT. We thank ESA for OC-CCI chlorophyll, Copernicus Marine Environmental Monitoring Services (CMEMS) for SLA and NOAA for SST data. Ship time was provided under grant AWI_PS113_00.

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
