# Peer review of "High-resolution physical-biogeochemical structure of a filament and an eddy of upwelled water off Northwest Africa"

_Ocean Science, 2019_

## Referee Comment (RC1) · Anonymous Referee #1 · 13 Dec 2019

This article presents the results of high resolution observations made in May 2018 with a towed instrument (Triaxus). Satellite observations are used to obtain a synoptic image to interpret the results. An almost straight, almost meridional section is realized offshore Cape Blanc (Mauritania). Water samples and CTD casts were collected. Surface currents were measured from VMADCP and a X-band marine radar. In situ temperature and salinity are measured. Concentrations of biogechemistry elements such as nitrate, phosphate, silicate, particulate organic carbon, phytoplankton and chlorophyll-a of seven major phytoplankton groups are obtained.

The section went through a mesoscale anticyclonic eddy and an upwelling filament. A

special focus is also given to a frontal region where symmetric instability could have been a source of nutriments in the euphotic layer. This is a good example of interdisciplinary research in the context of coastal upwelling.

This manuscript is clear and well written. observations and results are clearly explained. This article deserves publication with minor revision.

Specific comments:

Line 217 "VMADCP and the WaMoS radar (green/magenta in Figure 2b) agree well with the geostrophic flow" This is actually not the case, they are significant differences in magnitudes and directions throughout the section. This should be discussed in more details for the different processes. Since WaMoS radar current observation is a promising method, a proper assesment against VMADCP is valuable.

Figure 2: A high resolution infrared and/or ocean color image should provide a much better synoptic view of the eddy and upwelling filament.

Line 270: "This large scale (order of 50 km) coherent flow structure" - this is of the order of the Rossby radius (40-50 km) so this is not large scale, but mesoscale.

Line 322: "filament that came from the west" rather from the east, doesn't it ?

Line 345 "envrionmental"
* * *

---

## Referee Comment (RC2) · Anonymous Referee #2 · 14 Jan 2020

Review of os-2019-108 High-resolution physical-biogeochemical structure of a filament and an eddy of upwelled water off Northwest Africa

This well-written manuscript focuses on a high-resolution (O(1 km) horizontal, O(1 m) vertical) transect offshore of the upwelling region of northwest Africa, utilizing a towed undulating vehicle. It is a diagnostic study of the physical and biogeochemical structure, along the transect in relation to remotely sensed observations of an upwelled offshore plume that intersects the transect, and mesoscale eddy features. The physical structure measured in situ includes the plume, and mesoscale and submesoscale eddies. The biogeochemical structure, in relation to the physical structure, is diagnosed to arise

in various places from the nutrient enriched plume of upwelled water, symmetric instability, eddy trapping of water and subduction of low PV waters. Although the transect itself is well measured, the authors lack the observations (e.g., time series and across-transect gradients – except for remotely sensed data) to make definitive diagnoses of the observed physical-biogeochemical structures, with many uses of qualifying words and phrases such as "may", "suggests that", etc., in section 3.3 and onwards. However, they do make clear that 1-D dynamical and biogeochemical assumptions are very inadequate in this region, and by extension others like it, for analyzing water column profile data. They also illustrate the value of high resolution physical and biogeochemical data that towed undulating vehicles, such as the instrumented Triaxus-E can provide For example, they show how the observations suggest that observed high nitrate values in a narrow section of the mixed layer are consistent with submesoscale eddies at the base of the mixed later having symmetric instabilities injecting higher nutrient waters up into the mixed layer in a slantwise fashion. Such observations are very inefficient with traditional profiling instrument packages, which limits the collection of such highly resolved data sets over longer transects. The authors suggest that more highly resolved data sets like this need to be collected in eastern boundary upwelling regions to better understand/constrain atmospheric-ocean fluxes carbon dioxide.

I am puzzled by some of the information in the paragraph beginning on line 236. I think the authors are arguing that the + nitrate excess in the core of the anticyclone is due to 1) upwelled water with high nitrate, low DO (high AOU), and zero nitrate excess; 2) the residence time in the ML before the water is subducted in the eddy formation is $\sim$ weeks; 3) the residence time is long enough for air-sea flux to raise DO (lower AOU), but too short for PP to lower nitrate and hence leads to + nitrate excess. I am not an expert on this subject but time scales of O(weeks) seem long enough for PP to be significant. Further down in the manuscript it is mentioned that the phytoplankton doubling time is 1-2 days. Or am I misunderstanding the argument?

Minor comments Line 92. I suggest that "...we describe the used data." be replaced

with "...we describe the data used." or simply "...we describe the data."

Line 125. I am a little confused about the statement of identical gradients traversed in opposite directions by the saw tooth pattern. Is the idea that the vertical gradient is essentially the same in the downcast as the upcast?

Line 236. I suggest changing "...small scale aspects that could be resolved at much higher resolution than by traditional CTS casts ..." with "...small scale aspects that were resolved at much higher resolution than would have been by traditional CTS casts ..."

Line 419. Replace "lead" with "leads"

––––––––––––––––––––––––––––––––

---

## Author Comment (AC1) · 17 Jan 2020

We thank the reviewer for their very positive general assessment of the paper and for the specific suggestions that improved the quality of the paper.

Please note that we also added an additional piece of information in section 2 (old line 96) on the technical setup which we had forgotten before: "The Triaxus flew a so-called saw-tooth pattern (Figure 1a) and was slightly deflected to the side by yaw flaps so as not to measure in the ship's wake."

Comment:

[Figure]

Line 217 "VMADCP and the WaMoS radar (green/magenta in Figure 2b) agree well with the geostrophic flow" This is actually not the case, they are significant differences in magnitudes and directions throughout the section. This should be discussed in more details for the different processes. Since WaMoS radar current observation is a promising method, a proper assessment against VMADCP is valuable.

Response:

We agree that VMADCP and WaMoS current data require a proper and documented comparison/assessment. In fact, two of the authors of the current manuscript were involved in exactly such an assessment based on data from the same cruise as presented here: Hessner, K., El Naggar, S., von Appen, W.-J., and Strass, V. H.: On the reliability of surface current measurements by X-band marine radar, Remote Sensing, 11, https://doi.org/10.3390/rs11091030, 2019. We had cited this paper which was published in early 2019 in the data section (old line 172), but had not mentioned its scope properly. We thank the reviewer for raising this point and address it by referring to Hessner et al on (old) line 217: "(We refer to Hessner et al. (2019) for a more complete assessment of the differences between WaMoS and VMADCP data; they document a good qualitative and quantitative agreement for all data from the same cruise as discussed here.)"

Comment:

Figure 2: A high resolution infrared and/or ocean color image should provide a much better synoptic view of the eddy and upwelling filament.

Response:

We had looked at this in the past, but due to cloud cover, there were no such high resolution data sets available within a few days of the transect. However, upon reexamining, we realized that the OCCCI chlorophyll concentration 8-day average from two weeks after our transect contains useful information which we agree adds useful context to

the paper. We have added this as a new Figure 3 and have added its data description and interpretation as follows:

(new) line 213: "The Ocean Colour Climate Change Initiative (OC-CCI) chlorophyll-a dataset, Version 3.1, of the European Space Agency (Sathyendranath et al., 2018) was downloaded from https://esa-oceancolour-cci.org. These are 8-day composites of merged sensor (MERIS, MODIS Aqua, SeaWiFS LAC and GAC, VIIRS) products."

(new) line 243: "High resolution SST and chlorophyll concentration (from ocean color) data could provide a much better synoptic view of the (sub-)mesoscale features discussed in this paper. Those sensors are affected by cloud cover and therefore no such high resolution data is available in the study area within a few days of our 22.5 hour long transect. However, the average from 10- Jun-2018 to 17-Jun-2018 (Figure 3) which is approximately two weeks after the Triaxus transect contains sufficient information to be interpretable. The recently upwelled cold water near the coast is high in chlorophyll. The mesoscale filament providing the westward advection pathway is also associated with elevated chlorophyll concentrations. At the longitude of the transect, it has moved slightly southwards from approximately 20.9°N (e.g. Figures 4b/7d discussed below) to 20.4°N within the two weeks between the transect and the available satellite derived chlorophyll map (Figure 3). It is also clear that the two anti-cyclones (closed blue contours) are associated with very low chlorophyll concentrations."

(new) Figure 3 caption: "Map of satellite based chlorophyll concentration [$\mu$/l; OC-CCI v3.1 product] as color with sea level anomaly [m; SSALTO/DUACS SSH L4] as blue contours. The chlorophyll is an 8-day average centered on 13-Jun-2018 12:00 UTC and the SLA is a 7-day average centered on 13-Jun-2018 00:00 UTC, i.e. approximately two weeks after the Triaxus tow. The ship track and the westward advection pathway are plotted as in Figure 2."

Addition to acknowledgements: "We thank ESA for OC-CCI chlorophyll, Copernicus Marine Environmental Monitoring Services (CMEMS) for SLA and NOAA for SST data."

Comment:

Line 270: "This large scale (order of 50 km) coherent flow structure" - this is of the order of the Rossby radius (40-50 km) so this is not large scale, but mesoscale.

Response:

Yes, we agree and replaced "large scale" with "mesoscale".

Comment:

Line 322: "filament that came from the west" rather from the east, doesn't it ?

Response:

Yes. Thanks for catching our typo. Replaced "west" with "east".

Comment:

Line 345 "envrionmental"

Response:

Typo corrected.
* * *
[Figure]

**Fig. 1.** Added figure in response to comment by reviewer #1

---

## Author Comment (AC2) · 17 Jan 2020

We thank the reviewer for their very positive general assessment of the paper and for the specific suggestions that improved the quality of the paper.

Please note that we also added an additional piece of information in section 2 (old line 96) on the technical setup which we had forgotten before: "The Triaxus flew a so-called saw-tooth pattern (Figure 1a) and was slightly deflected to the side by yaw flaps so as not to measure in the ship's wake."

Comment:

[Figure]

I am puzzled by some of the information in the paragraph beginning on line 236. I think the authors are arguing that the + nitrate excess in the core of the anticyclone is due to 1) upwelled water with high nitrate, low DO (high AOU), and zero nitrate excess; 2) the residence time in the ML before the water is subducted in the eddy formation is âĹij weeks; 3) the residence time is long enough for air-sea flux to raise DO (lower AOU), but too short for PP to lower nitrate and hence leads to + nitrate excess. I am not an expert on this subject but time scales of O(weeks) seem long enough for PP to be significant. Further down in the manuscript it is mentioned that the phytoplankton doubling time is 1-2 days. Or am I misunderstanding the argument?

Response:

We assume that your comment regarded the paragraph beginning on (old) line 321 (not line 236).

We appreciate that the reviewer made us think through the argument again. We think that the argument about the time scale over which primary production acts was an inadvertent red herring. We have removed it from the discussion on the nitrate excess and instead we clarified the role of gas exchange:

Added sentences: "Assuming that the correlation in Figure 7c corresponds to the near Redfield behavior of phytoplankton growth and remineralization in the study area, primary production and remineralization acting in a parcel of water can move it parallel to the correlation line (black line in Figure 7c), but not change its nitrate excess (i.e. move it further away from the line). A change in the nitrate excess can only be achieved by a process that acts differently upon nitrate than upon oxygen. Gas exchange with the atmosphere likely plays that role as it only acts upon oxygen, but not upon nitrate. The gas exchange rate also depends upon how far from equilibrium with the atmosphere (100% oxygen concentration) the parcel of water is. That is, the gas exchange rate will be faster upon upwelling of the extremely undersaturated water thereby pumping oxygen into the water from the atmosphere. This corresponds to a reduction of AOU and
a corresponding increase in the nitrate excess. After primary production has acted, the water may be close to saturation or slightly super-saturated. However, because the water is closer to equilibrium then, the loss of oxygen to the atmosphere will be comparatively slower. Thus a nitrate excess can be present in the mixed layer relatively soon after upwelling (as measured by the relatively slow gas exchange near equilibrium). If the water gets subducted (stopping the gas exchange) during that time, a nitrate excess can also be observed later on."

Delete sentences: "At the same time, the residence time in the upper ocean probably was not long enough for primary production to utilize available nitrate before the water was subducted. We hypothesize that the water in the core of the anti-cyclone during our section spent only a short time (âĹijweeks) in the mixed layer before being subducted (i.e. removed from equilibrating processes in the mixed layer) to form the anti-cyclone." and "In a similar sense, the filament south of 21°N must have been part of an offshore advection "highway" from the upwelling region whose advective time was short enough for the above argument to hold."

Comment:

Line 92. I suggest that ". . .we describe the used data." be replaced with ". . .we describe the data used." or simply ". . .we describe the data."

Response:

Changed to "we describe the data".

Comment:

Line 125. I am a little confused about the statement of identical gradients traversed in opposite directions by the saw tooth pattern. Is the idea that the vertical gradient is essentially the same in the downcast as the upcast?

Response:

Yes, that is the idea, but we agree that it was not clear enough in the original text. Thus we elaborated this point in more detail: "Near the upper and lower turning points of the saw tooth pattern (see Figure 1), the vertical property gradients in the ocean are traversed in opposite directions (upwards and downwards) within short periods of time. Since the gradients (at least on average) will not have changed over that time period, the upcast and downcast data should show the same. For temperature, this is roughly the case, but for other sensors it is not due to their sensor lag. Attempting to shift the sensor data in time with respect to temperature such that the gradients, on average, become identical allows for a determination of the sensor lag."

Comment:

Line 236. I suggest changing ". . .small scale aspects that could be resolved at much higher resolution than by traditional CTS casts ..." with "...small scale aspects that were resolved at much higher resolution than would have been by traditional CTS casts ..."

Response:

We changed the wording similar to your suggestion to "small scale aspects that were resolved at much higher resolution than would have been possible with traditional CTD casts or with satellite observations."

Comment:

Line 419. Replace "lead" with "leads"

Response:

Thanks for pointing out this problem, but we meant to use the past tense (see preceding and succeeding sentences) of "to lead", not its present "leads", hence we replaced it with "led".

---

## Author Response (AR2)

Topic Editor Decision: Publish subject to minor revisions (review by editor) (23 Jan 2020) by Piers Chapman
Comments to the Author:
The authors have done a good job on responding to the comments of the reviewers and the paper is essentially ready for publication. However, there are a few specific items that need correcting.

Specific comments:

1. Line 34: Should be "depleted" not "deplete."
2. Lines 65-66: While the authors corrected the vertical displacement of cyclonic and anti-cyclonic eddies in line 64 following my earlier comment, they did not change the following lines. I believe lines 65-66 should read "…eddies that contain downward displaced isopycnals below upward displaced ones." rather than as written. Otherwise stratification will increase, not decrease. This then agrees with lines 78-79.
3. Line 107: "configuration used" not "used configuration."
4. Line 128: Perhaps you could change "should show the same" to "be identical."
5. Line 156: "only available during day time."
6. Lines 161-162: Do you really need the information here about the use of the instrument in sea ice conditions? The sentence could be finished after "can be tracked exactly." In fact, the whole paragraph could be omitted if you do not discuss any of the measurements mentioned here.
Line 166: Presumably this should read something like "with a grid spacing of 5m vertically and 0.02° (~2.2 km) horizontally respectively."

Response:
We thank the topic editor for his careful reading and have implemented all the suggested changes.

[revised manuscript text omitted]